

# Genome-wide identification and characterization of non-specific lipid transfer proteins in cabbage

Jialei Ji*, Honghao Lv*, Limei Yang, Zhiyuan Fang, Mu Zhuang, Yangyong Zhang, Yumei Liu and Zhansheng Li

Key Laboratory of Biology and Genetic Improvement of Horticultural Crops, Ministry of Agriculture/Institute of Vegetables and Flowers, Chinese Academy of Agricultural Sciences, Beijing, China
* These authors contributed equally to this work.

## ABSTRACT

Plant non-specific lipid transfer proteins (nsLTPs) are a group of small, secreted proteins that can reversibly bind and transport hydrophobic molecules. NsLTPs play an important role in plant development and resistance to stress. To date, little is known about the nsLTP family in cabbage. In this study, a total of 89 nsLTP genes were identified via comprehensive research on the cabbage genome. These cabbage nsLTPs were classified into six types (1, 2, C, D, E and G). The gene structure, physical and chemical characteristics, homology, conserved motifs, subcellular localization, tertiary structure and phylogeny of the cabbage nsLTPs were comprehensively investigated. Spatial expression analysis revealed that most of the identified nsLTP genes were positively expressed in cabbage, and many of them exhibited patterns of differential and tissue-specific expression. The expression patterns of the nsLTP genes in response to biotic and abiotic stresses were also investigated. Numerous nsLTP genes in cabbage were found to be related to the resistance to stress. Moreover, the expression patterns of some nsLTP paralogs in cabbage showed evident divergence. This study promotes the understanding of nsLTPs characteristics in cabbage and lays the foundation for further functional studies investigating cabbage nsLTPs.

Corresponding author
Limei Yang, ylmcaas@163.com

## INTRODUCTION

Non-specific lipid transfer proteins (nsLTPs), which are involved in binding and transporting various lipids, widely exist in the plant kingdom (*Edstam et al., 2011*). All known plant nsLTPs precursors include an N-terminal signal peptide, indicating nsLTPs are secreted proteins (*Carvalho & Gomes, 2007*). The mature nsLTPs are small proteins characterized by an eight-cysteine motif (8CM) with the basic form of C-Xn-C-Xn-CC-Xn-CXC-Xn-C-Xn-C (*José-Estanyol, Gomis-Rüth & Puigdomènech, 2004*). These eight cysteines are engaged in four disulfide bonds that stabilize the three-dimensional structure of the hydrophobic cavity, which allows the binding of different lipids and hydrophobic compounds (*Douliez et al., 2000*; *Salminen, Blomqvist & Edqvist, 2016*).

Based on sequence similarity, the nsLTPs from *Arabidopsis thaliana*, rice, Solanaceae, and other plants are divided into 10 types (I, II, III, IV, V, VI, VII, VIII, IX and X) (*Boutrot, Chantret & Gautier, 2008*; *Liu et al., 2010*; *Li et al., 2014*). Recently, plant nsLTPs have been categorized into four major and several minor types by intron position, sequence identity and spacing between the cysteine residues in the 8CM, as well as the post-translational modifications (*Edstam et al., 2011*; *Salminen, Blomqvist & Edqvist, 2016*).

Plant nsLTPs are involved in multiple physiological functions, such as transport of cuticular lipids, cutin synthesis, cell wall extension, pollen development, pollen tube growth and guidance, stigma and pollen adhesion, plant signaling, biotic stresses, abiotic stresses and seed maturation (*Kader, 1996*; *Kiełbowicz-Matuk, Rey & Rorat, 2008*; *DeBono et al., 2009*; *Nieuwland et al., 2005*; *Safi et al., 2015*). In addition, some plant nsLTPs have been identified as relevant allergens in plant foods and pollens (*Liu et al., 2010*). Moreover, plant nsLTPs display a complex tissue-specific and developmental expression pattern. NsLTPs are mainly expressed in the pericarp, tapetum, epidermal cells of embryos, stems, leaves and roots (*Zhang et al., 2008*; *DeBono et al., 2009*; *Edstam et al., 2013*). Interestingly, the expression of some *nsLTPs* can be induced by biotic and abiotic stresses, such as low or high temperature, drought, heavy metal exposure and disease (*Jung, Kim & Hwang, 2003*; *Gorjanović et al., 2004*; *Wu et al., 2004*; *Sun et al., 2008*; *Saltzmann et al., 2010*; *Guo et al., 2013*; *Schweiger et al., 2013*; *Yu et al., 2014*). The overexpression of a barley nsLTP gene in transgenic Arabidopsis enhanced the tolerance of Arabidopsis to bacterial pathogens (*Molina & García-Olmedo, 1997*). In addition, overexpression of a pepper nsLTP gene (CALTP1) in transgenic Arabidopsis could enhance the resistance of Arabidopsis against infection by *Pseudomonas syringae* pv. *tomato* and *Botrytis cinerea* and the tolerance of Arabidopsis to NaCl and drought stresses at various vegetative growth stages (*Jung, Kim & Hwang, 2005*).

Cabbage (*Brassica oleracea* var. *capitata*) is an economically important vegetable which is an important source of vitamin C, vitamin K and other phytochemicals, such as sulforaphane and indole-3-carbinol (*Wu et al., 2010*; *Ji et al., 2018*). So far, information on cabbage nsLTP genes is lacking. The whole-genome sequence of *B. oleracea* var. *capitata* was completed in 2013 (*Liu et al., 2014*). Based on genome-wide analysis, 89 putative nsLTP-encoding genes in cabbage were identified. These genes were classified into six types (1, 2, C, D, E and G). The gene structures, physical and chemical characteristics, motifs, subcellular localization and evolutionary patterns of the nsLTP family were investigated. Furthermore, we analyzed the expression patterns of nsLTP genes in various cabbage organs. The expression patterns of the nsLTP genes in response to biotic and abiotic stresses were also investigated. These results lay a foundation for further analyses of nsLTPs functions and provide a framework for the utilization of nsLTP-encoding genes to breed cabbage with an increased quality and stress resistance.

# MATERIALS AND METHODS

## Identification of cabbage nsLTP genes

The entire cabbage proteome was downloaded from the *B. oleracea* database (Bolbase, http://www.ocri-genomics.org/bolbase). Proteins with the HMMPfam domain PF00234

(protease inhibitor/seed storage/LTP family) in the cabbage proteome were identified by using the software platform HMMER3 (*Eddy, 2011*). Moreover, BLASTP was also used to identify the cabbage nsLTPs. All known *A. thaliana* nsLTP sequences were obtained from The *Arabidopsis* information resource (http://www.arabidopsis.org). These *Arabidopsis* sequences were then used as queries in a search against the cabbage protein database in Bolbase using BLASTP with the *e*-value ≤1*e*-3. The newly identified cabbage nsLTP amino acid sequences were assessed via an analysis of the 8CM backbone. The proteins lacking essential cysteines were discarded.

## Amino-acid sequence analysis

The signal peptide cleavage sites in the candidate nsLTP precursors were analyzed by using the SignalP 4.1 program (*Petersen et al., 2011*). The proteins without N-terminal signal sequences (NSSs) were removed. The *Arabidopsis* protease inhibitor and storage protein sequences were used in a comparative analysis with the rest of the candidate nsLTPs to identify and discard potential protease inhibitors and storage proteins. The C-terminal glycosylphosphatidylinositol-anchored (GPI-anchored) signals in the candidate nsLTP proteins were analyzed with the big-PI Plant Predictor program (*Eisenhaber et al., 2004*). The subcellular localization of nsLTP was predicted by TargetP 1.1 and Plant-PLoc server. After removing the signal peptide, the theoretical isoelectric point (pI) and molecular weight (Mw) of mature nsLTP proteins were calculated by the compute pI/Mw tool provided in EXPASY (http://web.expasy.org/compute_pi/). MEME software (v4.12.0) was used to search for motifs in all 89 cabbage nsLTP proteins, with a motif window length from 6 to 50 bp (*Bailey et al., 2009*). The three-dimensional structures of the cabbage nsLTPs were constructed and analyzed by Phyre2 and PyMOL (*Kelley et al., 2015*).

## Orthologous analysis and chromosome localization

The Arabidopsis nsLTP gene and protein sequences were used as queries in a BLAST search against the cabbage genome and proteome database, with a coverage of ≥0.75 and an *e*-value of ≤1*e*-10. The syntenic orthologous genes in cabbage and *Arabidopsis* were identified based on gene collinearity and sequence similarity (*e*-value ≤1*e*-20). All nsLTP genes were mapped to the cabbage chromosomes based on the genome information obtained from Bolbase. The chromosome localization map was made by using MapInspect software (http://mapinspect.software.informer.com/).

## Alignment and phylogenetic analysis of nsLTPs

The nsLTP proteins of *Marchantia polymorpha*, *Physcomitrella patens*, *Selaginella moellendorffii*, *Adiantum capillus-veneris*, *Pinus taeda*, *Oryza sativa*, *Arabidopsis thaliana* and *B. oleracea* were aligned with MAFFT (v7.037) (*Katoh & Standley, 2013*). Phylogenetic trees were constructed using FastTree (http://www.microbesonline.org/fasttree/) based on the WAG+CAT model (*Price, Dehal & Arkin, 2009*).

## RNA-seq data and bioinformatic analysis

The expression levels of the nsLTP genes in seven different cabbage organs (root, callus, leaf, stem, bud, flower and silique) were investigated using RNA-seq data under the

accession number GSE42891 in the Gene Expression Omnibus database. The expression patterns of the nsLTP genes in response to low and high temperatures were analyzed based on the RNA-seq data (ID: NN-0259-000003, NN-0259-000004, NN-0252-000006 and NN-0252-000003) from the National Agricultural Biotechnology Information Center (http://nabic.rda.go.kr/). An analysis of nsLTP gene expression in response to black rot disease in disease-resistant and disease-susceptible cabbage was carried out based on RNA-seq data (SRA098802) from the NCBI Sequence Read Archive (SRA) database. RNA-seq data (SRP091687) from male sterile/fertile buds was downloaded from the SRA database.

After the data processing of raw sequences, clean reads were aligned against the *B. oleracea* genome (http://www.ocri-genomics.org/bolbase/) using Tophat (v2.0.12). The transcript abundance of nsLTP genes was calculated by fragments per kilobase of exon model per million mapped reads (FPKM) or reads per kilobase per million mapped reads. The differential gene expression analysis was performed using DESeq (v1.16). Genes with FDR-adjusted $P$-value < 0.05 and fold-changes >2 were identified as differentially expressed genes (DEGs).

## RNA extraction and subcellular localization

Total RNA from cabbage (line 02–12) leaves was isolated using the RNAprep Pure Plant Kit (TIANGEN, Beijing, China) according to the manufacturer's instructions. Total RNA was reverse-transcribed using the PrimeScript™ RT reagent kit (TaKaRa, Kyoto, Japan). The full coding sequences of *Bol014756* (*BoLTP1.7*) and *Bol021902* (*BoLTP2.3*) were PCR-amplified with the primers PF1, PR1, PF2 and PR2 (Table S1) and inserted into a pBWA(V)HS-GFP vector, resulting in an N-terminal fusion with GFP under the control of the constitutive CaMV35S promoter. The fusion constructs were introduced into tobacco leaf epidermis as previously described (Sparkes et al., 2006). The fluorescence signals were detected using the confocal laser-scanning microscope (Nikon C1, Tokyo, Japan).

# RESULTS

## Identification of the putative nsLTP genes in cabbage

The nsLTP genes were identified by using the HMM search program in HMMER3 and a BLAST search against cabbage proteome. Initially, a total of 135 proteins with the conserved HMMPfam domain PF00234 were retrieved. The cysteine residue patterns of these protein sequences were then analyzed. Fifteen proteins lacking the essential cysteine residues were omitted. After that, 10 proteins lacking NSSs were also excluded (Table S2). Moreover, 12 proteins similar to protease inhibitors or storage proteins and nine hybrid proline-rich proteins were also discarded (Table S2). As a result, a total of 89 nsLTPs, designated BoLTPs in this study, were identified in the cabbage genome (Table 1). Based on the orthology analysis, 74 BoLTP genes were found to have orthologous relationships with 42 *A. thaliana* nsLTP genes (Table 1). Among these orthologous genes, 49 BoLTP genes were syntenic orthologues of 27 *A. thaliana* nsLTP genes (Fig. 1A).

**Table 1 The occurrence of non-specific lipid transfer proteins in cabbage and some of their features.**

| Name[a] | Signal peptide | | Mature protein | | | Intron amount/position[c] | Orthologous genes in *A. thaliana* |
|---------|----------------|---------|----------------|-----------|------|---------------------------|------------------------------------|
|         | Amino acids | Target[b] | Amino acids | Mass (Da) | pI   |                           |                                    |
| **Type 1** | | | | | | | |
| BoLTP1.1  | 24 | s | 95  | 9,922.37  | 7.40  | 1 (5) | AT3G51590 |
| BoLTP1.2  | 25 | s | 91  | 9,394.83  | 10.13 | 0     | AT3G51600 |
| BoLTP1.3  | 24 | s | 92  | 9,310.59  | 7.71  | 1 (5) | AT2G18370 |
| BoLTP1.4  | 25 | s | 91  | 9,682.25  | 11.42 | 0     | AT3G51600 |
| BoLTP1.5  | 19 | s | 92  | 9,549.78  | 8.51  | 0     | AT3G08770 |
| BoLTP1.6  | 25 | s | 91  | 9,693.23  | 11.27 | 0     | AT3G51590 |
| BoLTP1.7  | 24 | s | 90  | 9,498.97  | 8.19  | 0     | AT3G51590 |
| BoLTP1.8  | 23 | s | 92  | 9,175.57  | 8.49  | 1 (5) | AT5G59320 |
| BoLTP1.9  | 23 | s | 87  | 8,723.11  | 8.70  | 0     | AT5G59310 |
| BoLTP1.10 | 26 | s | 136 | 14,404.74 | 9.13  | 1 (5) | AT4G33355 |
| BoLTP1.11 | 18 | s | 116 | 11,905.88 | 9.37  | 0     | AT2G38540 |
| BoLTP1.12 | 22 | s | 96  | 10,212.54 | 3.63  | 0     | AT5G62065 |
| BoLTP1.13 | 25 | s | 94  | 9,590.03  | 9.08  | 0     | AT2G38530 |
| BoLTP1.14 | 25 | s | 94  | 9,590.03  | 9.08  | 0     | AT2G38530 |
| BoLTP1.15 | 25 | s | 94  | 9,520.92  | 8.91  | 0     | AT2G38540 |
| BoLTP1.16 | 23 | s | 100 | 10,493.99 | 8.70  | 1 (5) | AT5G59320 |
| BoLTP1.17 | 23 | s | 87  | 8,792.32  | 9.03  | 0     |           |
| BoLTP1.18 | 21 | s | 97  | 10,061.41 | 7.69  | 1 (5) | AT4G33355 |
| BoLTP1.19 | 20 | s | 93  | 9,648.84  | 8.73  | 0     | AT3G08770 |
| **Type 2** | | | | | | | |
| BoLTP2.1  | 28 | s | 68  | 7,411.84  | 9.75  | 0     | AT3G18280 |
| BoLTP2.2  | 20 | s | 76  | 7,716.91  | 8.20  | 0     | AT1G66850 |
| BoLTP2.3  | 29 | s | 68  | 7,143.28  | 8.90  | 0     | AT1G48750 |
| BoLTP2.4  | 28 | s | 68  | 7,381.78  | 9.54  | 0     | AT3G18280 |
| BoLTP2.5  | 24 | s | 73  | 7,877.10  | 4.80  | 0     | AT3G57310 |
| BoLTP2.6  | 30 | s | 67  | 7,636.89  | 8.17  | 0     | AT1G73780 |
| BoLTP2.7  | 28 | s | 68  | 7,460.87  | 9.94  | 0     | AT3G18280 |
| BoLTP2.8  | 30 | s | 68  | 7,645.89  | 8.42  | 0     | AT1G73780 |
| BoLTP2.9  | 24 | s | 78  | 8,524.93  | 5.18  | 0     | AT5G38195 |
| BoLTP2.10 | 24 | s | 75  | 7,792.09  | 7.44  | 0     |           |
| BoLTP2.11 | 29 | s | 68  | 7,255.47  | 9.06  | 0     | AT1G48750 |
| BoLTP2.12 | 21 | s | 74  | 7,647.76  | 7.71  | 0     | AT1G66850 |
| **Type C** | | | | | | | |
| BoLTPc1   | 34 | s | 64  | 6,729.72  | 6.71  | 0     | AT5G52160 |
| **Type D** | | | | | | | |
| BoLTPd1   | 23 | s | 78  | 8,281.64  | 8.19  | 0     |           |
| BoLTPd2   | 19 | s | 89  | 9,521.31  | 9.15  | 1 (4) | AT4G30880 |
| BoLTPd3   | 28 | s | 87  | 9,234.97  | 9.30  | 0     |           |

(Continued)

| Name[a] | Signal peptide | | Mature protein | | | Intron amount/position[c] | Orthologous genes in *A. thaliana* |
|---|---|---|---|---|---|---|---|
| | Amino acids | Target[b] | Amino acids | Mass (Da) | pI | | |
| BoLTPd4 | 28 | s | 86 | 9,285.70 | 8.86 | 1 (4) | AT4G33550 |
| BoLTPd5 | 23 | s | 92 | 9,521.52 | 10.48 | 1 (4) | AT2G37870 |
| BoLTPd6 | 19 | s | 90 | 9,607.40 | 8.71 | 1 (4) | AT4G30880 |
| BoLTPd7 | 28 | s | 84 | 8,910.50 | 8.72 | 0 | AT5G55450 |
| BoLTPd8 | 28 | s | 87 | 9,204.80 | 8.50 | 0 | AT5G55450 |
| BoLTPd9 | 28 | s | 90 | 9,616.54 | 9.06 | 1 (4) | AT5G55450 |
| BoLTPd10 | 23 | s | 78 | 8,348.74 | 8.68 | 0 | |
| BoLTPd11 | 23 | s | 92 | 9,517.49 | 10.42 | 1 (4) | AT2G37870 |
| BoLTPd12 | 26 | s | 73 | 7,445.50 | 4.46 | 0 | AT5G55450 |
| BoLTPd13 | 30 | s | 89 | 9,707.56 | 8.80 | 0 | AT5G55450 |
| BoLTPd14 | 26 | s | 73 | 7,521.59 | 4.46 | 0 | |
| BoLTPd15 | 27 | s | 76 | 8,139.38 | 5.04 | 0 | AT5G48490 |
| BoLTPd16 | 28 | s | 79 | 8,381.89 | 9.03 | 1 (4) | AT5G55410 |
| BoLTPd17 | 28 | s | 74 | 7,756.12 | 8.88 | 0 | AT5G55410 |
| BoLTPd18 | 23 | s | 91 | 9,356.81 | 8.70 | 1 (4) | AT3G53980 |
| **Type E** | | | | | | | |
| BoLTPe1 | 24 | c | 97 | 10,408.26 | 4.72 | 0 | AT3G52130 |
| BoLTPe2 | 25 | s | 97 | 10,290.02 | 4.72 | 0 | AT3G52130 |
| **Type G** | | | | | | | |
| BoLTPg1 | 21 | s | 173 | 17,684.24 | 5.64 | 1 (4) | AT1G27950 |
| BoLTPg2 | 22 | s | 185 | 18,970.72 | 8.45 | 2 (1, 155) | AT4G08670 |
| BoLTPg3 | 22 | c | 181 | 18,590.24 | 7.70 | 2 (1, 134) | |
| BoLTPg4 | 21 | s | 174 | 17,641.27 | 6.11 | 1 (4) | AT1G27950 |
| BoLTPg5 | 17 | s | 173 | 17,386.80 | 4.31 | 2 (4, 926) | AT3G43720 |
| BoLTPg6 | 19 | s | 285 | 29,369.39 | 4.70 | 2 (1, 147) | AT1G36150 |
| BoLTPg7 | 19 | s | 160 | 15,708.96 | 8.14 | 2 (19, 167) | AT1G18280 |
| BoLTPg8 | 22 | s | 136 | 13,860.91 | 5.59 | 1 (19) | AT1G18280 |
| BoLTPg9 | 22 | s | 160 | 15,697.94 | 7.73 | 2 (19, 167) | AT1G18280 |
| BoLTPg10 | 31 | s | 157 | 16,414.03 | 4.68 | 1 (19) | AT4G22666 |
| BoLTPg11 | 25 | s | 124 | 12,877.69 | 6.30 | 2 (4, 126) | AT1G62790 |
| BoLTPg12 | 24 | s | 147 | 14,871.51 | 5.05 | 2 (7, 140) | AT3G22600 |
| BoLTPg13 | 20 | s | 155 | 15,572.72 | 6.07 | 2 (−5, 40) | AT1G18280 |
| BoLTPg14 | 19 | s | 190 | 19,568.39 | 5.19 | 1 (1) | AT1G36150 |
| BoLTPg15 | 26 | s | 178 | 18,208.64 | 4.34 | 4 (4, 164, 371, 617) | AT1G05450 |
| BoLTPg16 | 25 | s | 154 | 15,227.71 | 7.11 | 2 (4, 211) | |
| BoLTPg17 | 24 | s | 148 | 14,812.25 | 8.48 | 2 (4, 289) | |
| BoLTPg18 | 22 | s | 90 | 9,701.35 | 6.99 | 1 (13) | AT1G73560 |
| BoLTPg19 | 21 | s | 116 | 12,333.19 | 5.42 | 1 (7) | AT1G73550 |
| BoLTPg20 | 23 | s | 182 | 18,472.01 | 4.99 | 1 (31) | AT3G22620 |

| Name[a] | Signal peptide | | Mature protein | | | Intron amount/position[c] | Orthologous genes in *A. thaliana* |
|---|---|---|---|---|---|---|---|
| | Amino acids | Target[b] | Amino acids | Mass (Da) | pI | | |
| BoLTPg21 | 26 | s | 162 | 16,361.87 | 8.73 | 2 (7, 188) | AT1G73890 |
| BoLTPg22 | 22 | s | 158 | 15,803.06 | 6.93 | 2 (13, 173) | AT1G18280 |
| BoLTPg23 | 21 | s | 127 | 13,708.39 | 4.18 | 2 (13, 193) | AT1G73550 |
| BoLTPg24 | 22 | s | 127 | 13,220.35 | 5.62 | 2 (13, 163) | AT1G73560 |
| BoLTPg25 | 26 | s | 175 | 17,570.14 | 5.66 | 2 (4, 173, 410, 623) | AT1G05450 |
| BoLTPg26 | 22 | c | 179 | 18,303.98 | 8.45 | 2 (1, 131) | |
| BoLTPg27 | 21 | s | 174 | 17,769.26 | 6.90 | 1 (4) | AT1G27950 |
| BoLTPg28 | 27 | s | 147 | 14,350.26 | 4.69 | 2 (4, 126) | AT2G13820 |
| **Type X** | | | | | | | |
| BoLTPx1 | 25 | s | 99 | 10,667.26 | 4.57 | 0 | |
| BoLTPx2 | 22 | s | 95 | 10,049.59 | 8.15 | 0 | |
| BoLTPx3 | 23 | s | 118 | 13,635.59 | 5.76 | 1 (1) | AT1G52415 |
| BoLTPx4 | 24 | s | 76 | 8,355.81 | 8.70 | 0 | AT1G64235 |
| BoLTPx5 | 24 | s | 68 | 7,128.46 | 9.05 | 0 | AT1G64235 |
| BoLTPx6 | 22 | s | 92 | 10,068.84 | 8.68 | 0 | AT4G08530 |
| BoLTPx7 | 28 | s | 106 | 11,759.57 | 6.09 | 1 (12) | |
| BoLTPx8 | 24 | s | 96 | 10,034.94 | 7.67 | 0 | |
| BoLTPx9 | 21 | s | 101 | 10,653.27 | 7.71 | 0 | |

**Notes:**
[a] The accession numbers of BoLTPs are shown in Table S2.
[b] Subcellular target for the protein. S = secretory pathway; c = chloroplast.
[c] Position of the intron is given as the number of bases from the codon encoding the eighth cysteine in the 8CM.

## Classification of nsLTP genes and their distribution in chromosomes

The plant nsLTPs can be divided into four major and several minor types according to intron position, sequence identity and spacing between the Cys residues in the 8CM, as well as the post-translational modifications (*Edstam et al., 2011*; *Salminen, Blomqvist & Edqvist, 2016*). Based on the presence of a GPI modification site, the classification was initiated by first sorting the identified BoLTPs into Type G. In the second round of classification, the remaining BoLTPs were sorted based on the identity matrix calculated from the multiple sequence alignments. When compared with the classification proposed by *Edstam et al. (2011)* and *Salminen, Blomqvist & Edqvist (2016)*, we found that 80 out of the 89 BoLTPs could be categorized into six types (1, 2, C, D, E and G). Type1, type2, type D and type G nsLTPs, which encompassed 19, 12, 18 and 28 nsLTP genes, respectively, clearly represented a large proportion of the BoLTPs. Moreover, nine cabbage proteins displayed less than 30% identity with all other studied BoLTPs, so they were listed individually and named BoLTP×1 to BoLTP×9 (Table 1).

The chromosomal location of each *BoLTP* gene was confirmed based on the cabbage genome information in Bolbase. A total of 68 (76.4%) *BoLTP* genes were distributed across nine chromosomes, and the rest were located on the unanchored scaffolds (Fig. 1B).

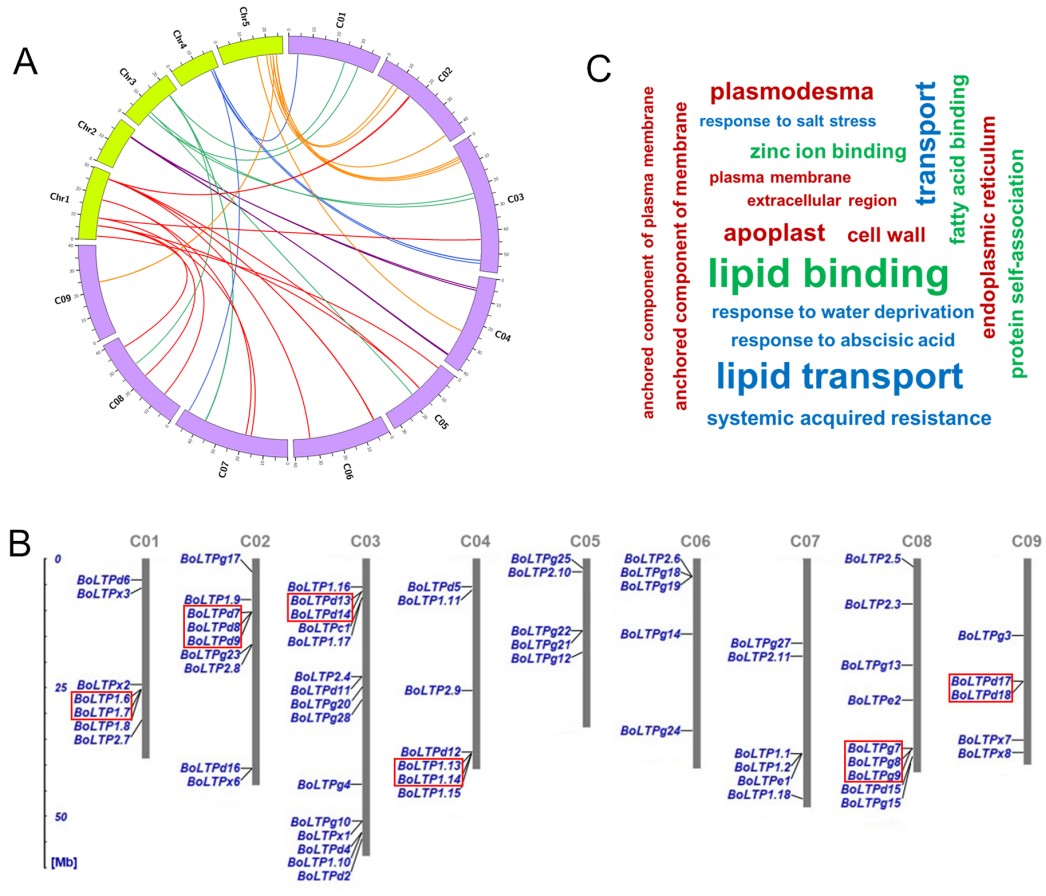

Figure 1 Circos diagram of syntenic nsLTP genes in *A. thaliana* and cabbage (A); genomic localization of the BoLTP genes in the chromosomes of cabbage (B); gene ontology categories of *BoLTPs* (C). Ch1–Ch5 are *A. thaliana* chromosomes, and C01–C10 are cabbage chromosomes. The tandem duplication repeats are indicated by red rectangles.

Seven genes were located in chromosome 1 and 4, nine genes were located in chromosomes 2 and 8, 15 genes were located in chromosome 3, five genes were located in chromosome 5, 6 and 9, and six genes were located in chromosomes 7. In cabbage, seven direct repeat tandems consisting of 16 nsLTP genes were identified (Fig. 1B). One tandem of two duplicated nsLTP genes was presented in chromosome 1 (BoLTP1.6 and BoLTP1.7), chromosome 3 (BoLTPd13 and BoLTPd14), chromosome 4 (BoLTP1.13 and BoLTP1.14), chromosome 9 (BoLTPd17 and BoLTPd18) and Scaffold000118_P2 (BoLTP×4 and BoLTP×5). One tandem of three duplicated nsLTP genes was present in chromosome 2 (BoLTPd7, BoLTPd8 and BoLTPd9) and chromosome 8 (BoLTPg7, BoLTPg8 and BoLTPg9).

## Characteristics of cabbage nsLTPs

The characteristics of the 89 BoLTPs are summarized in Table 1. All the BoLTP protein precursors possess a signal peptide of 17–34 amino acids. The putative subcellular localization of BoLTPs was analyzed. As expected, most of the proteins are predicted to be secreted except for BoLTPe1, BoLTPg3 and BoLTPg26, which have been predicted to be

**Table 2 Some characteristics of the different types of non-specific lipid transfer proteins found in cabbage.**

| Type | GPI-anchored | Spacing pattern | | | | | | | | | |
|---|---|---|---|---|---|---|---|---|---|---|---|
| 1 | No | C | $X_9$ | C | $X_{13,14,16}$ | CC | $X_{19}$ | CXC | $X_{19,21,22,23,24}$ | C | $X_{13}$ | C |
| 2 | No | C | $X_7$ | C | $X_{13}$ | CC | $X_8$ | CXC | $X_{23}$ | C | $X_{5,6}$ | C |
| C | No | C | $X_9$ | C | $X_{16}$ | CC | $X_9$ | CXC | $X_{12}$ | C | $X_6$ | C |
| D | No | C | $X_{9,10,14}$ | C | $X_{14,15,16,17,19}$ | CC | $X_{9,11,12}$ | CXC | $X_{19,22,24}$ | C | $X_{6,7,8,9,10}$ | C |
| E | No | C | $X_{13}$ | C | $X_{15}$ | CC | $X_9$ | CXC | $X_{22}$ | C | $X_6$ | C |
| G | Yes | C | $X_{6,9,10}$ | C | $X_{11,13,14,16,17,18}$ | CC | $X_{12}$ | CXC | $X_{23,24,25,26,29}$ | C | $X_{5,7,8,9}$ | C |

**Note:**
Character "X" represents any amino acid, and the Arabic numeral following "X" stands for the numbers of amino acid esidues.

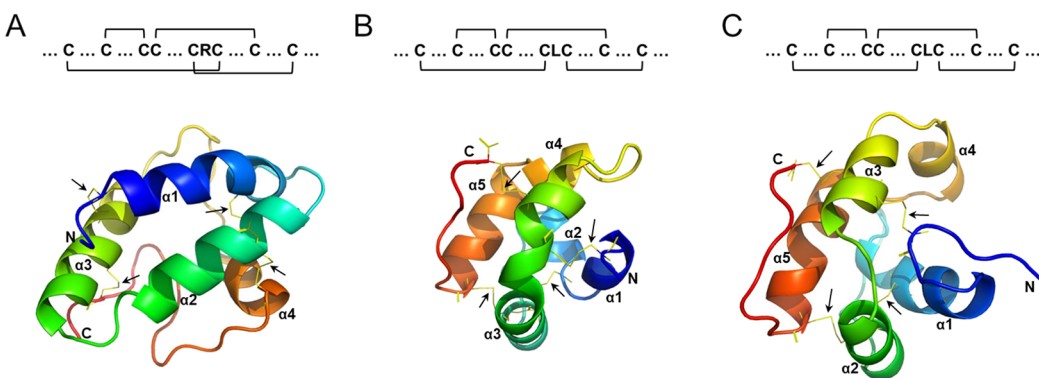

**Figure 2 Schematic representation of the structure and cystein pairing pattern of BoLTP1.5 (A), BoLTP2.1 (B) and BoLTPd15 (C).** The four disulfide bonds are shown in yellow.

chloroplast proteins (Table 1). Except for type G nsLTPs, the Mws of the mature BoLTPs usually range from 6,729 to 11,906 Da, indicating the cabbage nsLTPs genes encode small proteins. Because the mature nsLTPs of type G have more amino-acid residues in the C-terminus than the other mature BoLTP proteins, they have much higher Mws (range from 12,333.19 to 29,369.39 Da). Among the 89 BoLTPs, 57 display a basic pI (7.11–11.42) and the rest show an acidic pI (3.63–6.99). The gene ontology categories of *BoLTPs* are shown in Fig. 1C.

To further analyze the characteristics of the BoLTPs, a multiple sequence alignment of mature BoLTPs was conducted using MAFFT. Obviously, the 8CMs of the 89 BoLTPs are conservative (Fig. S1). Moreover, the amino acid sequence alignment of the 8CMs of BoLTPs reveals a variable number of inter-cysteine amino acid residues (Table 2). In order to better understand the ligand binding and tertiary structure of BoLTPs, BoLTP1.5, BoLTP2.1 and BoLTPd15 were selected as representative sequences of type 1, 2 and D for structural modelling. The structural analysis showed that BoLTP1.5 includes two conserved pentapeptides, T-P-V-D-R (positions 60–64) and P-Y-S-I-S (positions 100–104). It has been reported that these two consensus pentapeptides (T/S-X-X-D-R/K and P-Y-X-I-S) play an important role in catalysis or binding (*Douliez et al., 2000*). As shown in Fig. 2, the 3D structures of these three

BoLTPs were predicted and analyzed by Phyre2 and PyMOL. Each BoLTP possesses a compact α-helical domain consisting of four or five α-helices connected by short loops (Fig. 2). Four disulfide bonds formed by the eight-cysteine residues can stabilize the three-dimensional structure of the hydrophobic cavity. In BoLTP1.5, the Cys residues 1–6, 2–3, 4–7 and 5–8 are paired (Fig. 2A), whereas the Cys residues 1–5, 2–3, 4–7 and 6–8 are paired in BoLTP2.1 and BoLTPd15 (Figs. 2B and 2C). This phenomenon indicated that the central residue of the CXC motif may influence the Cys pairing and the overall fold of the protein. As shown in Fig. S1, the X position of CXC motif is hydrophilic in type 1 BoLTPs. However, X is a hydrophobic residue in the CXC motif of Type 2 and Type D BoLTPs. Except for the disulfide bonds, many H-bonds also engage in the stabilization of the three-dimensional structures of BoLTPs. The particular folding structure forms an internal tunnel-like cavity that can bind different hydrophobic molecules.

## Phylogenetic analysis of the nsLTP family

To analyze the phylogenetic relationship of the nsLTPs among *M. polymorpha*, *Physcomitrella patens*, *S. moellendorffii*, *Adiantum capillus-veneris*, *Pinus taeda*, *O. sativa*, *Arabidopsis thaliana* and *B. oleracea*, 187 nsLTPs from these eight species were analyzed (Table S3). These nsLTP sequences were aligned using MAFFT (v7.037). Subsequently, the approximately-maximum-likelihood phylogenetic tree was constructed from the multiple sequence alignment based on the WAG+CAT model. Previously, the plant nsLTPs have been divided into 10 types (*Edstam et al., 2011*). Based on comparison with the previous dataset, the six groups divided in this study were in agreement with the type 1, 2, C, D, E and G of nsLTPs. As shown in the phylogenetic tree, members in type 1, 2, C and E formed specific clades, suggesting that the nsLTPs in these types share a common ancestor (Fig. 3; Fig. S2). Although type D and G can be well distinguished from other types, they cannot formed separated clades (Fig. 3; Fig. S2). In particular the nsLTPs of type G were divided into four subtypes (named G1–G4) (Fig. 3). It was also worth noting that no type E nsLTP was found in monocotyledon plants, which may discarded these genes during the evolutionary divergence between monocots and dicots. Generally, *A. thaliana* and *B. oleracea* are closer to each other and more distantly related to other species in each group of the phylogenetic tree, indicating the closer relationship between *A. thaliana* and *B. oleracea*.

## Expression analysis of nsLTP genes

To explore the spatial expression patterns of the BoLTP genes, RNA-seq data from seven different organs (root, stem, leaf, callus, bud, flower and silique) were used for an expression analysis of *BoLTPs*. The expression level of each BoLTP gene was estimated by the FPKM value, and the genes with FPKM ≥1 were identified as truly expressed genes (*Yao et al., 2015*). In this study, 72 (81%) BoLTP genes were expressed in at least one of the seven organs (Fig. 4A; Table S4). Interestingly, several BoLTP genes, such as *BoLTP1.7*, *BoLTP1.10*, *BoLTP1.18*, *BoLTP2.2*, *BoLTP2.12* and *BoLTPx2*, were specifically expressed in the buds. However, these genes were significantly

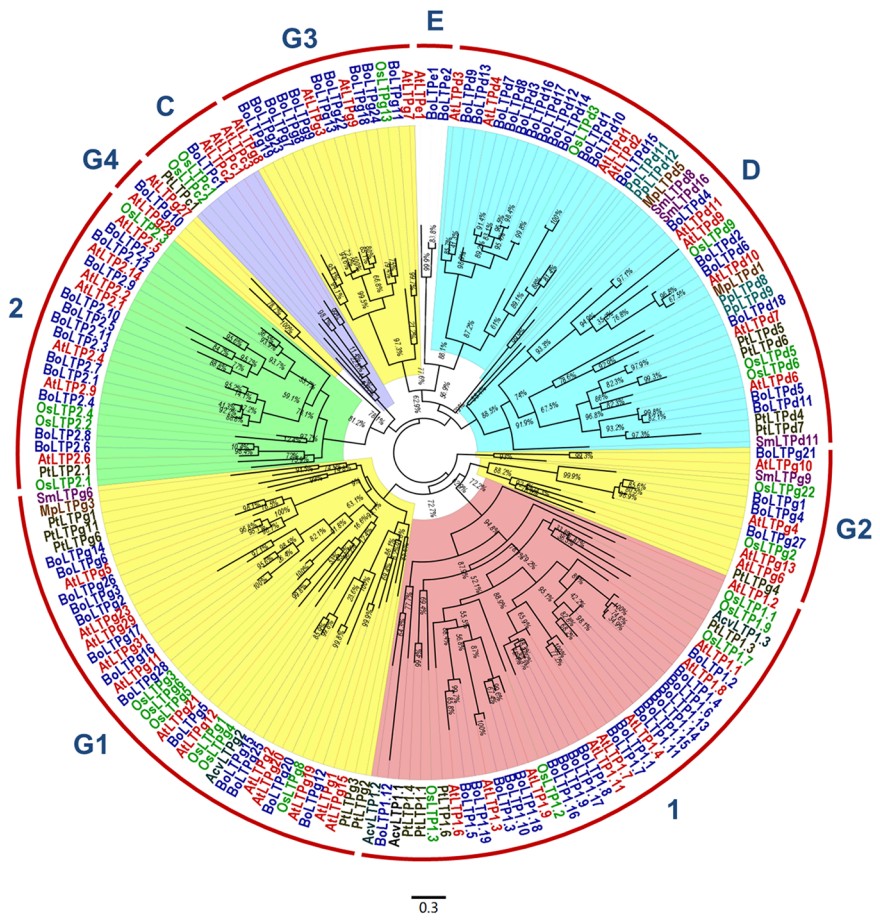

**Figure 3 Phylogenetic tree of the predicted cabbage nsLTP genes and the representative members of nsLTP gene family.** Phylogenetic analysis was performed using FastTree based on the WAG+CAT model. Different species are shown with different colors. The accession numbers of the nsLTPs are shown in Table S3.

downregulated in the buds of 83121A, which is a male sterile mutant with a defective exine (Fig. 4E; Table S5).

To analyze the relationship between BoLTPs and cabbage resistance to biotic and abiotic stress, comparative analyses of the expression of *BoLTPs* between the resistant and susceptible cabbage materials were conducted. Based on an analysis of RNA-seq data from cabbage treated with cold stress for 2 h, eight *BoLTPs* demonstrated significant changes between the cold-tolerant BN10600 and cold-susceptible BN10700. Of these eight DEGs, six were downregulated and two were upregulated in BN10700 (Fig. 4B; Table S5). Moreover, RNA-seq data from two cabbage lines, heat-tolerant BN1HS and heat-susceptible BN2HS, that had been treated by heat shock for 1 h, were used to explore the responses of BoLTP genes to high temperature. As shown in Fig. 4C, seven DEGs were identified between BN1HS and BN2HS (Table S5). Of these DEGs, five were downregulated and two were upregulated in BN2HS. Black rot caused by *Xanthomonas campestris* pv. *campestris* is a major disease of cabbage. In order to explore the black rot resistance genes in cabbage, the differential expression analysis between the

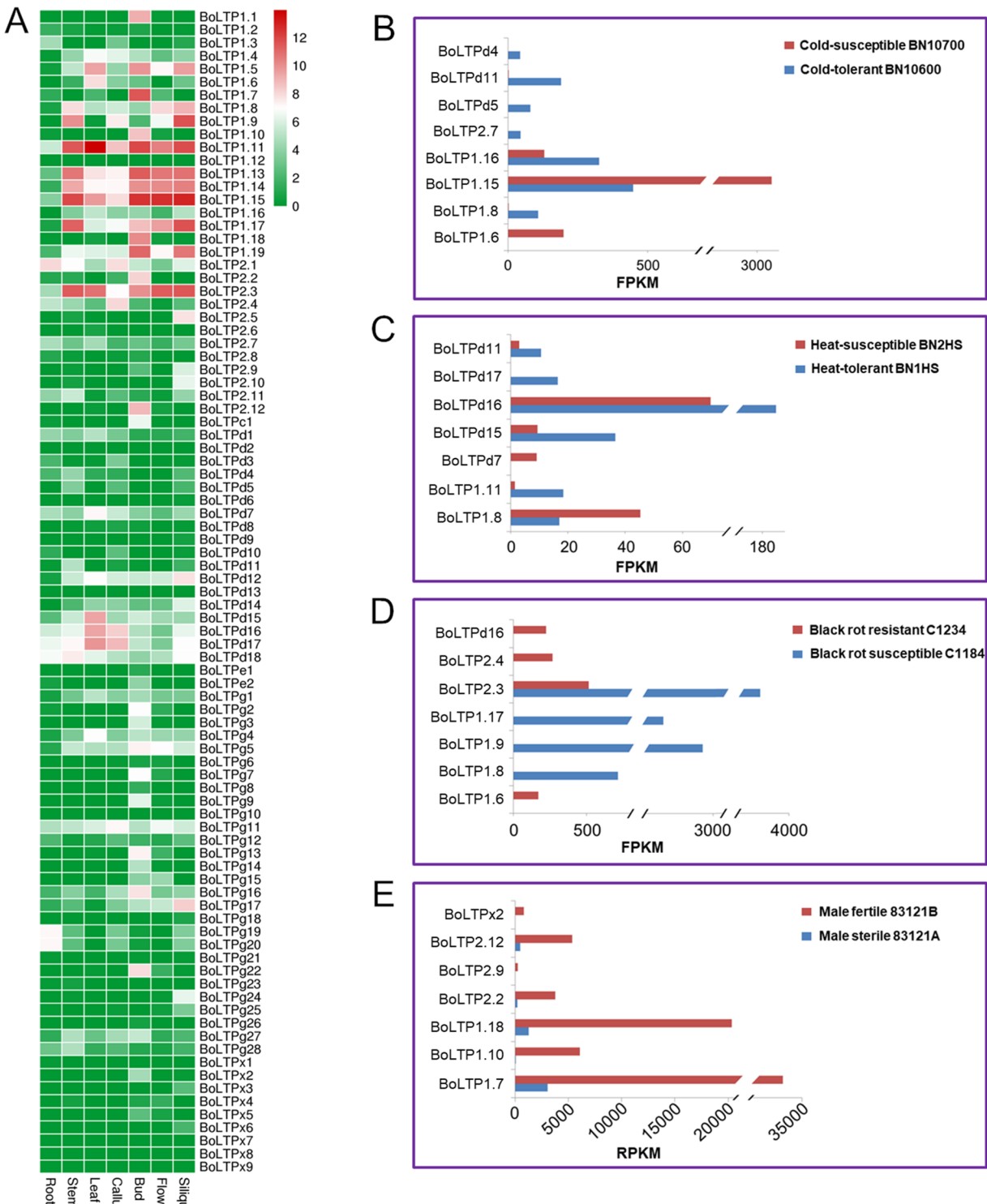

**Figure 4 Expression analysis of BoLTP genes.** (A) Expression profiles of BoLTP genes in seven organs. (B–D) Differential expression analysis of BoLTPs in response to cold, heat and black rot disease. (E) Differential expression analysis of BoLTPs in male sterile/fertile buds.

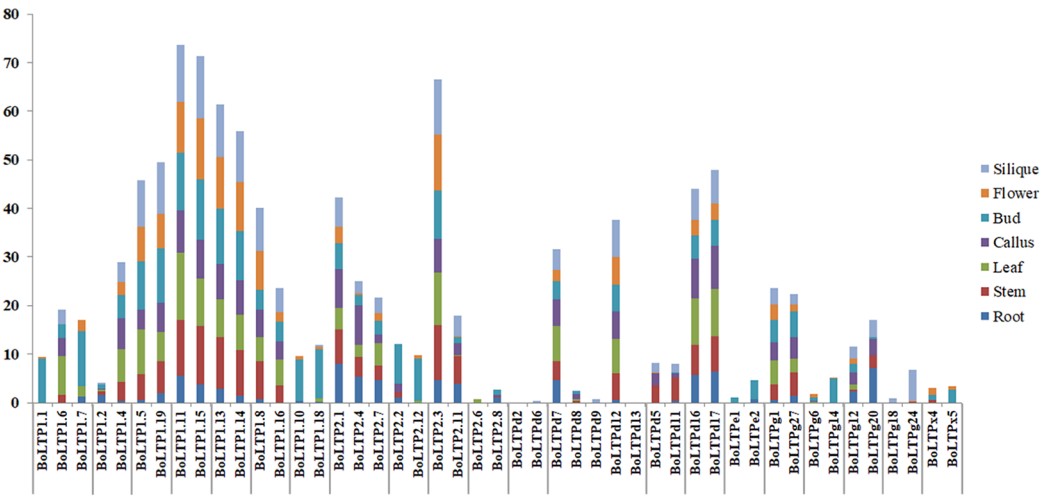

**Figure 5 Different transcript levels of duplicated BoLTP genes.** The BoLTP gene expression level was measured by $\log_2(1+FPKM)$ value.

two cabbage parental lines, C1234, which is black rot disease-resistant and C1184, which is a susceptible line, were also carried out. According to Fig. 4D, seven DEGs were identified between C1234 and C1184 (Table S5). Of these DEGs, three were downregulated and four were upregulated in C1184. Significantly, among these DEGs, four BoLTPs (1.6, 1.8, d2 and d11) DEGs may be related to the resistance to at least two types of stress.

The alteration of expression patterns is considered an important indicator of functional divergence between duplicated genes (*Makova & Li, 2003*; *Ganko, Meyers & Vision, 2007*; *Hellsten et al., 2007*). The differential expression analysis of the duplicated *BoLTPs* indicated significant expression differentiation among the paralogous *BoLTPs* (Fig. 5). For example, *BoLTP1.1*, *BoLTP1.6* and *BoLTP1.7* are orthologous to *AT3G51590*, which is known to be involved in sexual reproduction in *A. thaliana* (*Ariizumi et al., 2002*; *Chae et al., 2009*). *BoLTP1.1* and *BoLTP1.7* were mainly expressed in the floral organs, while *BoLTP1.6* was mainly expressed in the leaves and significantly upregulated in the black rot disease-resistant C1234 (Figs. 4D and 5). As another example, *BoLTPd7*, *BoLTPd8*, *BoLTPd9*, *BoLTPd12* and *BoLTPd13* are orthologous to *AT5G55450*, which is involved in disease resistance in *A. thaliana* (*McLaughlin et al., 2015*). *BoLTPd7* was expressed in every investigated organ, *BoLTPd8*, *BoLTPd9* and *BoLTPd13* were not expressed in any of the investigated organs, while *BoLTPd12* was expressed in every investigated organ except the root. Moreover, *BoLTPd7* and *BoLTPd8* responded to cold and heat stress, but their responses to the stresses were opposite (Figs. 4C and 4D). *BoLTPd9* responded to cold stress and was downregulated in the cold-susceptible BN10700 (Fig. 4C). These results indicate that the duplicated genes with different expression patterns may possess different physiological functions.

## DISCUSSION

In this study, 89 genes putatively encoding nsLTPs in cabbage were identified. Based on the classification system described by *Edstam et al. (2011)* and

*Salminen, Blomqvist & Edqvist (2016)*, these BoLTPs could be classified into six types (1, 2, C, D, E and G). Bioinformatic analysis predicted that most of BoLTPs are secreted proteins. In order to verify the predicted results, the bud-specific *BoLTP1.7* and black rot-responsive *BoLTP2.3* were selected as representative proteins for subcellular localization. The results showed that the fluorescence signals of BoLTP1.7 and BoLTP2.3 fused with GFP were detected in the extracellular environment, indicating they are secreted proteins (Fig. S3). Orthology analysis showed that most of the BoLTP genes have orthologous nsLTP genes in *A. thaliana*, indicating the cabbage nsLTP genes were derived from a common ancestor shared with *A. thaliana*. Compared with the Arabidopsis genes, the numbers of cabbage *nsLTPs* in type 1 and D were expanded, while the gene numbers of the other types were reduced or unchanged. It is worth mentioning that 39 Arabidopsis nsLTP genes did not have a BoLTP ortholog. These results indicate that there are not only gene duplications and triplications but also gene loss or mutation in the evolutionary process of cabbage.

Many studies have suggested that nsLTPs participate in sexual reproduction processes, such as pollen development, pollen exine formation and fertilization (*Ariizumi et al., 2002*; *Chae et al., 2009*; *Huang, Chen & Huang, 2013*; *Edstam & Edqvist, 2014*). Promoter activity analysis has suggested that *AT3G51590* (*LTP12*) is specifically expressed in tapetum at the uninucleate microspore stage and the bicellular pollen stage (*Ariizumi et al., 2002*). The type C *nsLTP* with exclusive expression in the tapetum, *AT5G52160*, has been shown contribute to pollen exine formation in Arabidopsis (*Huang, Chen & Huang, 2013*). In this investigation, a high number of BoLTP genes were found to be specifically or highly expressed in buds or flowers (Fig. 4A). Among these genes, *BoLTP1.1* and *BoLTP1.7* are orthologous to *AT3G51590*, and *BoLTPc1* is orthologous to *AT5G52160*. Moreover, *BoLTP1.7*, *BoLTP1.10*, *BoLTP1.18*, *BoLTP2.2*, *BoLTP2.12* and *BoLTPx2* were significantly downregulated in the buds of a male sterile mutant with defective exine. Notably, these nsLTP proteins were also significantly down-accumulated in the male sterile mutant buds (*Ji et al., 2018*). The results suggested that the *BoLTPs* specifically or highly expressed in floral organs may play an important role in the sexual reproduction progress in cabbage.

There is much evidence that nsLTPs are related to the resistance to various types of stresses, including the resistance to phytopathogens, freezing, drought and salt (*Molina & García-Olmedo, 1997*; *Hincha, 2002*; *Jung, Kim & Hwang, 2003*, *2005*; *McLaughlin et al., 2015*). Some Arabidopsis nsLTPs have been classified as pathogenesis-related PR-14 proteins, and most of them belong to the type 1 nsLTP group (*Sels et al., 2008*). In this study, we found that most of these pathogenesis-related Arabidopsis genes have *BoLTPs* orthologs in cabbage. According to recent studies, the paralogous genes *AT5G59310* (*AtLTP1.11*) and *AT5G59320* (*AtLTP1.12*) could negatively regulate plant immunity in Arabidopsis. The overexpression of *AT5G59320* (*AtLTP1.12*) enhances susceptibility of Arabidopsis to virulent bacteria and reduces the resistance of Arabidopsis to avirulent bacteria (*Gao et al., 2015*). In contrast, the double mutant *AtLTP1.11/AtLTP1.12* showed an increased resistance to *Pseudomonas* (*Gao et al., 2015*). Similar to the orthologous genes *AtLTP1.11* and *AtLTP1.12*, *BoLTP1.8* and *BoLTP1.9* showed high expression levels in

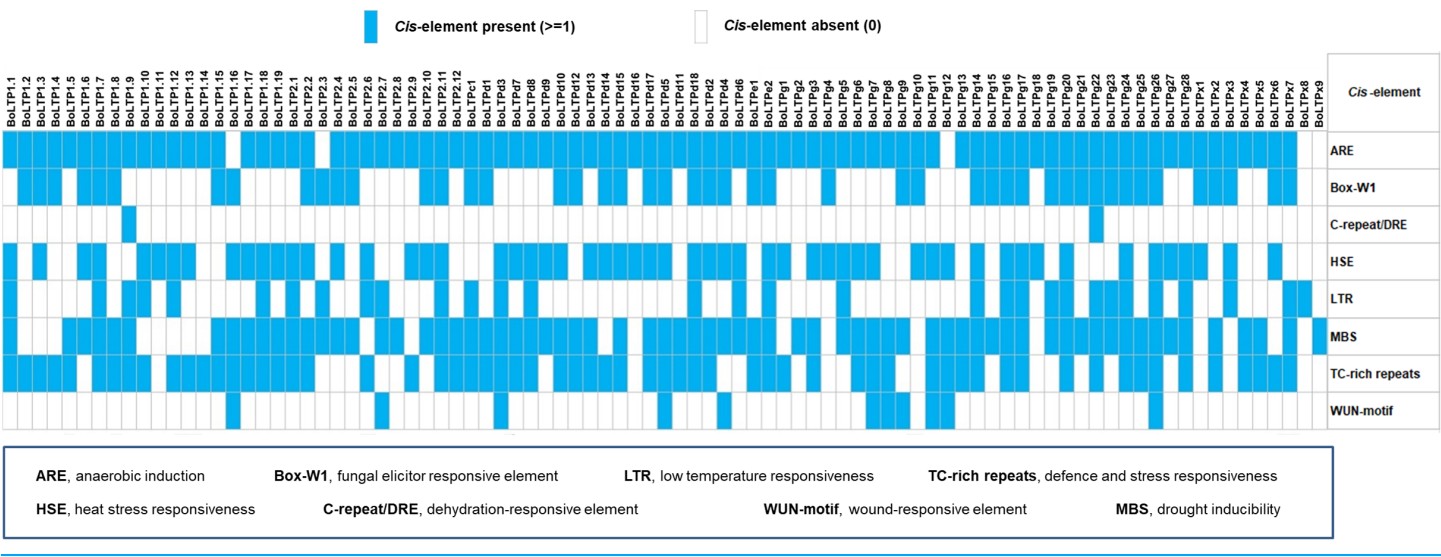

**Figure 6 Summary of stress-related cis-elements in the upstream regions of BoLTP genes.** The putative cis-elements in the promoters were analyzed using PlantCARE.

black rot-susceptible C1184 and undetectable expression levels in black rot-resistant C1234, indicating that the high expression of these two genes may contribute to black rot disease susceptibility in cabbage. Furthermore, an analysis of stress-relevant cis-elements in the promoter regions of the BoLTP genes was conducted. The results show that the promoters of the BoLTP genes possess at least one stress-related cis-element (Fig. 6), indicating that the BoLTPs are involved in the stress response. Given all that, the BoLTP genes identified in this study could be used to further seek stress-resistant genes in cabbage and other Brassica crops.

Moreover, the expression analyses among paralogs indicated that some duplicated BoLTP genes showed significantly different expression patterns (Fig. 5). Previous studies have shown that the alteration of expression patterns is an important indicator of functional divergence between duplicated genes (*Makova & Li, 2003*; *Ganko, Meyers & Vision, 2007*; *Hellsten et al., 2007*). In other words, functional divergence could eliminate the redundancy of these duplicated BoLTP genes, which may be beneficial in multiple biological processes, such as cabbage growth, sexual reproduction and resistance to stress. However, in-depth studies should be carried out to reveal the biological functions of the BoLTPs in the development of different organs and in resistance to stress.

## CONCLUSIONS

A total of 89 *BoLTPs* were identified based on genome-wide research. These genes were classified into six different types (1, 2, C, D, E and G). The tertiary structure, phylogenetic development, and gene expression of the *BoLTPs* were also summarized. The expression analysis shows the functional importance of BoLTPs in sexual reproduction and stress response. It is important to continue to reveal the functions of these *BoLTPs* with basic experiments such as overexpression or knock-down strategies followed with detailed phenotypic investigations. Moreover, the *BoLTPs* identified in this

study provide a reference for creation of resistant materials or artificial regulation of cabbage fertility by using genetic engineering technology.

### Funding

This work was funded by the National Key Research and Development Program (2017YFD0101804), the Science and Technology Innovation Program of the Chinese Academy of Agricultural Sciences (CAAS-ASTIP-2013-IVFCAAS), the National High Technology Research and Development Program of China (863 Program, 2012AA100101), the Key Projects in the National Science and Technology Pillar Program during the Twelfth Five-Year Plan Period (2012BAD02B01), the Modern Agro-Industry Technology Research System (CARS-25-B-01), and the Project of Science and Technology Commission of Beijing Municipality (Z141105002314020-1). The funders had no role in study design, data collection and analysis, decision to publish, or preparation of the manuscript.

### Grant Disclosures

The following grant information was disclosed by the authors:
National Key Research and Development Program: 2017YFD0101804.
Science and Technology Innovation Program of the Chinese Academy of Agricultural Sciences: CAAS-ASTIP-2013-IVFCAAS.
National High Technology Research and Development Program of China: 863 Program, 2012AA100101.
National Science and Technology Pillar Program during the Twelfth Five-Year Plan Period: 2012BAD02B01.
Modern Agro-Industry Technology Research System: CARS-25-B-01.
Project of Science and Technology Commission of Beijing Municipality: Z141105002314020-1.

### Competing Interests

The authors declare that they have no competing interests.

### Author Contributions

- Jialei Ji conceived and designed the experiments, performed the experiments, analyzed the data, contributed reagents/materials/analysis tools, prepared figures and/or tables, authored or reviewed drafts of the paper, approved the final draft.
- Honghao Lv conceived and designed the experiments, performed the experiments, contributed reagents/materials/analysis tools, authored or reviewed drafts of the paper, approved the final draft.
- Limei Yang conceived and designed the experiments, analyzed the data, contributed reagents/materials/analysis tools, authored or reviewed drafts of the paper, approved the final draft.

- Zhiyuan Fang analyzed the data, contributed reagents/materials/analysis tools, authored or reviewed drafts of the paper, approved the final draft.
- Mu Zhuang analyzed the data, contributed reagents/materials/analysis tools, authored or reviewed drafts of the paper, approved the final draft.
- Yangyong Zhang analyzed the data, contributed reagents/materials/analysis tools, authored or reviewed drafts of the paper, approved the final draft.
- Yumei Liu contributed reagents/materials/analysis tools, authored or reviewed drafts of the paper, approved the final draft.
- Zhansheng Li contributed reagents/materials/analysis tools, authored or reviewed drafts of the paper, approved the final draft.

## Data Availability

The raw data are provided in the Supplemental Files.

## Supplemental Information

Supplemental information for this article can be found online at http://dx.doi.org/10.7717/peerj.5379#supplemental-information.

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
