# Peer review of "Genome-wide identification and characterization of non-specific lipid transfer proteins in cabbage"

_PeerJ, doi:10.7717/peerj.5379_

## Round 0.1 · original submission · Major Revisions

The reviewers have made some very helpful comments, but reviewer 2 had some significant concerns which I agree with. Please ensure you address all these concerns

Reviewer 1 ·

Basic reporting

no comment

Experimental design

no comment

Validity of the findings

no comment

Additional comments

The manuscript” Genome-wide identification and characterization of non-specific lipid transfer proteins in cabbage” by Ji et al detailed analyzed non-specific lipid transfer proteins (nsLtps) in cabbage based on the public genome and transcriptome data and experimental analysis. This manuscript generally fit the criteria of PeerJ journal, clearly descript the work of nsLtps identification and related study. The experimental design was good. Research question well defined, relevant and meaningful. All the data the authors got were relatively reliable. However, there are still some problems need to be concerned and corrected:
1. In the part of characterization on 62 BonsLtps are summarized, authors didn’t explain why they chose BonsLtpI.7 and BonsLtpII.3 to perform the subcellular localization. The subcellular localization of All BonsLtps should be predicted by using bioinformatic tools (i.e. Cell-PLoc) and then select some interesting BonsLtps to do the subcellular localization.
2. Phylogenetic analysis of the nsLtp family revealed sLtps in Arabidopsis and cabbage are closer to each other. As we know, both Arabidopsis and cabbage are belonged to dicot, while rice is belonged to monocot. Is this difference due to the difference between dicot and monocot? Authors should combine more organisms nsLtp in phylogenetic analysis to get more biological meaning.
3. In the part of the expression analysis of nsLtp genes, authors combined public RNA-seq date to analysis the expression profiling of BonsLtps at the transcriptional level. Did they validate the RNA-seq data before analysis? That’s essential to make data more reliable.
Based on these concerns, I think this manuscript should be resubmitted with minor revisions.

Reviewer 2 ·

Basic reporting

In the introduction there are some errors or misunderstandings regarding the evolution, expression, function and lipid-binding of LTPs that needs correction.

• Firstly, for an update on plant LTPs I suggest that authors read and include a citation to Salminen et al 2016 (Planta. 244(5):971-997) which is a recent and very informative review on the plant LTPs.
• On line 2, when the authors write that the LTPs exists widely in the plant kingdom. The sentence should be complemented with a citation to Edstam et al 2011 (Mol Plant. 4(6):947-64). In this paper it is shown that LTPs are present in bryophytes, ferns, conifers and flowering plants.
• On line 33 the authors state that LTPs are involved in transfer of lipids between membranes. This is not correct since the LTPs are generally secreted and located on the outside of the plasma membrane.
• Also on line 33, the authors state that LTPs are transporting glycolipids. To my knowledge there is no publication supporting this statement. I suggest that authors provide a citation or rewrite the sentence.
• On lines 43-44, the authors mention that LTPs are involved in cell wall extension. This should be complemented with citation to Nieuwland et al (Plant Cell. 2005, 17(7):2009-19).
• On lines 43-45, it should be added that LTPs are suggested to be involved in transport of cuticular lipids. (Since this is proposed in DeBono et al 2009, which is cited.)
• On lines 49-50 the authors should mention that LTPs also are expressed in roots. (See for instance, Edstam et al. 2013 Plant Mol Biol. 83(6):625-49.)
• Lines 50-52, the authors should add some more recent citations regarding the stess-induced expression of LTPs.
• Line 56, the authors write that cabbage is cultivated due to its excellent quality. Here I suggest that the authors are more specific and describe which qualities they are referring to.

Comments on the reporting of the results
• Lines 243-250: Here the authors describe the expression pattern of Arabidopsis LTPs presented in other reports. This information should be moved to introduction or discussion.

Comments on discussion
• Lines 270-278 are irrelevant and should be omitted.
• Lines 291-292; include a citation to Edstam and Edqvist 2014 (Physiol Plant. 152(1):32-42) in which it is shown that several LTP genes are involved in maturation of seed and pollen.
• The discussion includes rather large segments where the results are repeated, for instance on lines 280-287, 314-319. I would suggest that these parts are omitted or minimized, and that that the discussion is used to put the obtained data in a larger context.

Experimental design

Line 82: Why were the sequences with GPI-anchor removed? Provide arguments:
Lines 111-113: The localization study should be performed also with 1-2 control proteins with known localization patterns, for instance in plasma membrane or cell wall. As it is the localization study seems rather meaningless. Also add information about the microscope used.
Lines 121: Why was the expression of LTPs during response to black rot disease investigated? Are there some specific arguments that support a role for LTPs during the response?
Line 148: I would recommend using the more recent LTP classification system described in Edstam et al 2011 (Mol Plant. 4(6):947-64) and Salminen et al 2016 (Planta. 244(5):971-997). This classification system is useful since it is possible to apply also on LTPs from bryophytes and lycopods, as well as on the LTPs from flowering plants. It is also less complicated as it divides the LTPs in fewer major types, and keeps the “traditional” types LTP1 and LTP2.

Validity of the findings

Lines 332-334: The authors should avoid listing experimental results in the conclusion. I would suggesting rewriting the conclusions with more focus on extensions of the work. But without using standard clichés (such as now on lines 337-338).

Additional comments

This manuscript is basically just an identification of the members of the LTP gene family. It is not providing us with any new clues to help us understand these enigmatic proteins. The information on the expression of LTPs in the male sterile cabbage is interesting and should be investigate further.

Unfortunately, the text in the introduction of the manuscript suggests that the authors lack detailed and updated knowledge of the plant LTPs.

---

## Round 0.2 · accepted · Accept

Many thanks for making the required changes to the manuscript

# Reviewer 1 ·

Basic reporting

1. Work description is Clear and unambiguous, professional English used throughout;
2. The introduction is sufficient, Authors detailed introduced the background and the work fits into the broader field of knowledge. Literatures referenced appropriately;
3. The structure of this article is acceptable with well-designed figures and tables;

Experimental design

Well designed experiments. Research question well defined, relevant and meaningful. All the data the authors got were relatively reliable.

Validity of the findings

no comment

Additional comments

The second version of the manuscript “Genome-wide identification and characterization of non-specific lipid transfer proteins in cabbage”, authors have made some revisions to respond the reviewers’ comments and upgraded figures to make data more robust. Based on this revised version, the manuscript meets the PeerJ criteria and should be accepted as is.

Reviewer 2 ·

Basic reporting

The manuscript is clear and easy to follow. However, the spelling should be checked once more.

Experimental design

No comments

Validity of the findings

No comments

Additional comments

The revised manuscript is very much improved compared to the original version. My previous remarks were carefully considered by the authors. I have no further comments.